# Intraoperative Administration of Adipose Stromal Vascular Fraction Does Not Improve Functional Outcomes in Young Patients with Anterior Cruciate Ligament Reconstruction

**DOI:** 10.3390/jcm11216240

**Published:** 2022-10-22

**Authors:** Wen-Chin Su, Ho-Yi Tuan-Mu, Hung-Maan Lee, Ting-Yu Hung, Kuan-Lin Liu

**Affiliations:** 1Department of Orthopedics, Hualien Tzu Chi Hospital, Buddhist Tzu Chi Medical Foundation, Hualien 970473, Taiwan; 2Department of Physical Therapy, College of Medicine, Tzu Chi University, Hualien 97004, Taiwan; 3Sports Medicine Center, Hualien Tzu Chi Hospital, Buddhist Tzu Chi Medical Foundation, Hualien 970473, Taiwan; 4School of Medicine, Tzu Chi University, Hualien 97004, Taiwan

**Keywords:** biologic augmentation, adipose stromal vascular fraction, functional outcomes, anterior cruciate ligament reconstruction

## Abstract

Adipose stromal vascular fraction (SVF) has a versatile cellular system for biologically augmented therapies. However, there have been no clinical studies investigating the benefits of the augmentation of anterior cruciate ligament reconstruction (ACLR) with SVF. We conducted a retrospective study in assessing the effects of intraoperative SVF administration on the functional outcomes in young patients with ACLR. The enrolled patients were divided into the control group (ACLR only) and the SVF group (ACLR with SVF). The functional outcomes in both groups were assessed by the Lysholm knee scoring system, the Tegner activity scale, and the International Knee Documentation Committee (IKDC) subjective evaluation form, and compared at several time points during a 12-month follow-up. We found that the sex distribution and pre-surgery scores were similar in the two groups, whereas the mean age of the SVF group was higher than that of the control group (*p* = 0.046). The between-group analysis and generalized estimating equation model analysis revealed that, while patients in the SVF group significantly improved all their functional outcomes at 12 months after surgery, this improvement was not significantly different from the results of patients in the control group (Lysholm, *p =* 0.553; Tegner, *p =* 0.197; IKDC, *p =* 0.486). No side effects were observed in either group. We concluded that the intraoperative administration of SVF does not improve or accelerate functional recovery after ACLR in young patients.

## 1. Introduction

Injury to the anterior cruciate ligament (ACL) is one of the most frequent injuries to the knee [1]. ACL ruptures lead to knee pain, instability, and impairment of daily living activities, all of which may result in prolonged periods of absence from work or sport [2]. ACL reconstruction (ACLR) remains the gold standard of treatment for ACL ruptures in young adults and athletes [3]. Although ACLR has a satisfactory success rate (73–95%), a significant number of patients cannot return to their pre-injury level of sport [4,5]. Additionally, because of the low healing capacity of ACL, it has been advocated that a post-surgery period of up to 12 m should be allowed in facilitating the biological recovery of the joint [6]. To improve the results, recent advances in the area of regenerative medicine have led to the emergence of novel biologically augmented ACLR techniques [1,5]. However, the clinical studies investigating ACLR combined with platelet-derived growth factor, various types of stem cells, or platelet-rich plasma (PRP) either yielded disappointing results [5,7,8,9] or provided a low degree of evidence [10,11,12]. Thus, the effectiveness of the augmentation of ACLR with other biological treatments warrants investigation.

Adipose stromal vascular fraction (SVF) is the aqueous fraction of adipose tissues, a heterogeneous, versatile cellular system including adipose-derived stem/stromal cells (ADSCs) [13,14,15]. SVF has several advantages over ADSCs such as easy preparation, no need for time-consuming ex vivo expansion, and containing other cell types with regenerative and healing potentials that involves paracrine activity modulation [15,16,17,18]. In orthopedic medicine [19], SVF has been used in treating patients with certain conditions, mainly degenerative cartilage diseases [19,20,21,22,23,24,25,26,27,28]. However, to date, there has been only one animal study reporting that SVF stimulated graft healing after ACLR [29]. Thus, clinical evidence is timely required in defining whether SVF has biologically augmented effects on ACLR.

In this retrospective study, we aimed to assess the effects of the intraoperative administration of SVF on functional outcomes in young patients with ACLR surgery. The enrolled patients were divided into two study groups: the control (non-SVF) group had ACLR only, and the SVF group had ACLR with SVF administration. The patient-reported functional outcomes in both study groups were assessed and compared at several time points during a 12-month follow-up.

## 2. Materials and Methods

### 2.1. Ethical Statement and Setting

This retrospective study was conducted at a medical center in Eastern Taiwan. The study was conducted according to the guidelines of the Declaration of Helsinki and approved by the Research Ethics Committee of Hualien Tzu Chi Hospital, Buddhist Tzu Chi Medical Foundation (approval number, No. IRB111-101-B). Informed consent was obtained from all the subjects involved in the study.

### 2.2. Study Design and Patients

Between 1 January 2020 and 31 December 2020, a total of 56 patients with ruptured ACLs attended our hospital and received arthroscopic ACLR. Patients were included if they (1) were 18 years or older and (2) were diagnosed with acute ACL rupture or irreparable ACL tear. Patients were excluded if they (1) could not communicate verbally or give responses to the questionnaires, (2) had ACLR before, (3) had a history of other knee problems, and (4) had pre-existing disorders that could influence their functional capacity. After exclusion, 35 patients were enrolled and divided into two study groups: the control non-SVF group (*n* = 16) only had ACLR surgery, and the SVF group (*n* = 19) had ACLR surgery with SVF augmentation (Figure 1). The SVF administration is a self-pay option, and the choice of this type of treatment was based on patients’ preferences after full communication. The pre-operative counseling for all patients included explanations of the scientific background of the potential benefits of SVF, the possible side effects of this treatment, and the self-pay amount of charge. After selecting the choice of SVF therapy, the patients received further consultations regarding the procedures of the treatment.

### 2.3. Surgical Procedures

In this study, a single surgeon performed ACLR in all patients with surgical procedures previously described [30]. In brief, standard anterolateral and anteromedial portals were established. The associated intra-articular injuries, such as meniscal ruptures and chondral lesions, were treated at the same time. The ACLR was performed under arthroscopic assistance with single bundle four-strand autologous hamstring tendon grafts. The femoral tunnel was set up first. Under deep knee flexion, an endoscopic femoral aimer with a 2 mm offset was placed by using an additional medial portal to the 10:30 or 1:30 o’clock position in the right or left knee, respectively. A tunnel was created according to the size of the graft and the loop length of EndoButton^®^ (Smith & Nephew Inc., Endoscopy Division, Andover, MA, USA) to keep at least 20 mm grafts in the femoral tunnel. The tibial tunnel was set up with the tibial aimer adjusted to a 45° position. The remnant was preserved in all cases. After setup, the four-strained autogenous tendon grafts suspended on the loop of the EndoButton^®^ were passed through the tunnel. At the tibial site, the grafts were fixed with a BIORCI^®^ (Smith & Nephew Inc., Endoscopy Division, Andover, MA, USA) interference screw. Additional cortical screws and washers were inserted to augment suture fixation.

### 2.4. SVF Preparation and Administration

For each patient in the SVF group, the autogenous adipose tissue was harvested by the same orthopedic surgeon before the ACLR procedures. Using the Coleman technique, the adipose tissue was harvested from the abdomen [31] and was managed by a single staffing nurse in the operation room. The methods for the preparation of SVF have been described previously [32]. Briefly, a fluid containing 250 cc normal saline, 1 mL epinephrine (1 mg/mL), and 20 mL 2% lidocaine was injected into the subcutaneous area of the abdomen through two punctured wounds 10 cm away from the umbilicus. After 15 min of waiting, liposuction was performed with a 3 mm multiport cannula. At least 80 cc of fat tissue was harvested. The harvested fat was allowed to stand for 10 min. The liquid portion was then discarded, and the adipose layer was collected. The fat was then processed via centrifugation at 1200× *g* for 3 min to generate Coleman fat. The Coleman fat was then mechanically emulsified by shifting between two 10 mL syringes connected by a female-to-female Luer lock adapter coupler with an internal diameter of 2.4 mm for a total of 40 times. After processing, the fat turned into an enmeshed fat, and it was then processed via centrifugation at 2000× *g* for three minutes. Finally, the sticky substance under the oil layer was extracted as the SVF/extracellular matrix gel (Figure 2A). After completing ACLR and eliminating the intra-articular fluid, the SVF gel was injected into the femoral tunnel, tibial tunnel, and intra-articular tendon substance following an arthroscopic guide (Figure 2B,C).

### 2.5. Rehabilitation Protocol

All patients were rehabilitated according to the same protocol. Immediate full weight-bearing was allowed, but the patients had to wear a functional range of motion brace (DonJoy X-Act ROM Knee Orthosis, Surrey, UK) during the first three months. The range of motion was restricted to 0–30 degrees in the first two weeks to avoid unexpected falls as a result of inhibition of the quadriceps. Then, the restriction was adjusted by increasing 20 degrees per week till the full range of motion was achieved. Physiotherapy for analgesia, as well as patella mobilization, progressive full range-of-motion exercises, and isometric quadricep contraction exercises, were allowed one-month post-surgery. Cycling was allowed at three months, and running was allowed at four months. A return to pivot sports was permitted at six months when the patient had regained full functional stability in terms of muscle strength, coordination, and balance, compared with the contralateral leg [33].

### 2.6. Functional Outcome Measurements

In our hospital, we routinely monitor patients’ functional outcomes after ACLR over time in evaluating their post-surgery recovery during the rehabilitation program. The patient-reported outcomes evaluated in this study included three types of questionnaires. The Lysholm knee scoring system [34] consists of eight items and is scored on a scale of 0 to 100; higher scores indicate fewer symptoms and higher levels of functioning. The Tegner activity scale [35] is a one-item score that graded activity based on work and sports activities on a scale of 0 to 10; zero represents disability because of knee problems, and 10 represents national or international level soccer. The IKDC subjective evaluation form [36] consists of 18 items and the summed score range from 0 to 100; higher scores represent lower levels of symptoms and higher levels of function and participation in sporting activity. All the patients completed these three types of questionnaires pre-surgery and at 2 weeks, 6 weeks, 3 months, 6 months, 9 months, and 12 months post-ACLR surgery. Data were collected at our outpatient clinic.

### 2.7. Sample Size Calculation

Sample size calculations were based on the difference in functional outcomes between the two study groups. Using a presumed effect size of 1.13 based on a previous study [23], a confidence level of 95%, and a power of 80%, the minimum sample size was estimated to be 14 patients per group.

### 2.8. Statistical Analysis

The Shapiro–Wilk test was used in checking the distribution of the continuous variables. The continuous variables were compared using an independent two-sample *t*-test in the cases of normal distribution of data or the Mann–Whitney U test in the cases of the non-normal distribution of data. All continuous variables are presented as mean ± standard deviation (SD) for the consistency of data presentation. A chi-square test was used in comparing the categorical variables which are presented as frequency and percentage. The generalized estimating equation (GEE) model was used in analyzing the association of variables (SVF treatment, sex, age, and follow-up time) with repeated-measure functional outcome scores after adjustment for the potential confounders. The Spearman rank correlation test was used in evaluating the correlation between any two functional outcome scores. A correlation coefficient of >0.7 indicated a strong correlation. A *p*-value < 0.05 was considered statistically significant. The statistical analysis was performed using the SPSS software program version 23.0 (Statistical Package of Social Sciences, Chicago, IL, USA) for Windows.

## 3. Results

### 3.1. Demographic Characteristics and Functional Outcomes

During the study period, 35 patients who received arthroscopic ACLR in our hospital were enrolled and were divided into the control (non-SVF) group (n = 16) and the SVF group (n = 19) (Figure 1). Between-group analysis revealed that the sex distribution in the SVF groups (female, n = 9 (47.4%); male, n = 10 (52.6%)) did not statistically differ from that in the non-SVF group (female, n = 7 (43.8%); male, n = 9 (56.2%); *p* = 0.830). However, the mean age of the SVF group (31.1 ± 10.0 years) was statistically higher than that of the non-SVF group (24.8 ± 7.3 years, *p* = 0.046). The functional outcomes were assessed using the Lysholm knee scoring system [34], the Tegner activity scale [35], and the International Knee Documentation Committee (IKDC) subjective evaluation forms [36]. Table 1 and Figure 3 show the functional outcome scores over time assessed by these three types of questionnaires in the two study groups. As shown, in both groups, each type of the functional outcome score immediately dropped at two weeks post-surgery, quickly returned to its pre-surgery level at six weeks post-surgery, and continued rising till the end of the study period to a level that was significantly higher than the pre-surgery value. The drops in the functional outcome score during the first two-week period were mainly due to pain and inactivity after surgery. The pre-surgery scores were similar in the two study groups. However, the between-group analysis revealed no significant differences in any of the functional outcome scores at any of the time points measured. Table 2 shows the correlation matrix of Spearman rank correlation coefficients. As shown, the correlation coefficients (>0.7) indicated that any of the two functional outcome scores had a strong correlation. No SVF-related serious adverse effects were observed in any of the patients.

### 3.2. GEE Model Analysis

The results from the GEE model analysis for the associations of variables with repeated-measure functional outcomes are shown in Table 3. As shown, after adjustment for potential confounders, the multivariate analysis revealed that the main effect of SVF on sex was not significant in any of these three functional outcome scores, whereas the main effect of age on follow-up time at 12 months post-surgery was significant.

## 4. Discussion

The most important finding of this study was that, while the patients receiving the intraoperative administration of SVF after ACLR significantly improved all their functional outcomes when assessed at 12 months after surgery, this improvement was not significantly different from the results of patients receiving ACLR without the SVF treatment. In these two study groups, the profiles of post-surgery recovery over the 12-month follow-up period were similar. Thus, the intraoperative administration of SVF as a therapeutic strategy does not improve or accelerate functional recovery after ACLR in young patients.

The clinical outcomes of ACLR continue to require significant improvement, especially for young adults and athletes who desire to return to pre-injury sporting activity [3]. The rate of the return to pre-injury activity level ranges from 37% to 75% [5]. In addition, graft healing after ACLR is very slow, leading to a long period of recovery time even with carefully monitored rehabilitation [5]. Indeed, a recovery period of 6–12 months in young patients and two years in athletes after ACLR surgery has been suggested [6]. To improve clinical outcomes, there has been a growing interest in the development of novel, biologically augmented ACLR techniques [1,5]. To date, among these techniques, the most widely used therapies in the clinical setting are ACLR combined with either stem cells or PRP but yielded unsatisfactory results or a low level of evidence [5,7,8,9,10,11,12]. For example, Nin et al. [7] reported that the use of platelet-derived growth factor in patients treated with bone-patellar tendon-bone allografts has no discernable clinical or biomechanical effect at 2 years of follow-up. Alentorn-Geli et al. [8] showed that patients receiving adipose-derived regenerative stem cells at the time of ACLR did not provide additional benefits regarding knee function and the healing/maturation of the graft at 12 months. In a recent meta-analysis including 14 studies, Zhu et al. [12] concluded that PRP applied alongside ACLR did not improve knee stability and the enlargement of tunnels and did not accelerate the healing of grafts. These two types of therapies mainly focus on the regenerative and healing properties of stem cells or platelets [9,10,11,12]. Recently, SVF has been proposed as a biological product superior to single-type cell therapies because it contains versatile cell types, including stem cells with therapeutic potential involving paracrine activity modulation, and preparing it for intraoperative application is easy [13,14,15,16,17,18]. In orthopedic medicine, SVF has been used in treating patients with various conditions, including knee osteoarthritis [20,21,22,23], chondromalacia [24], degenerative meniscal injuries [25], pseudoarthrosis [26], and Achilles tendinopathy [27,28], all of which show optimistic results. However, regarding the condition of the ACLR, there has been only one animal study. Using a rat model, Santoso et al. [29] reported that SVF stimulated graft healing after ACLR, as evidenced by histological assessments. To the best of our knowledge, this work is the first clinical study to investigate the biologic augmentation of ACLR with SVF.

In this study, the functional outcomes were routinely assessed during the follow-up period by using three types of questionnaires, all of which have been validated in patients with knee injuries [37]. The IKDC subjective evaluation form [36] assessed the domains of symptoms/pain, the functional activities of daily living, and function in sports and/or recreation. The first two domains are measured by the Lysholm knee scoring system [34]. The Tegner activity scale [35] gives a one-item score activity scale. Therefore, our assessments of the functional outcomes covered most of the domains of symptoms and activity that are specific to patients with ACLR. The patients’ responses to different questionnaires appeared to be very consistent because a strong correlation existed between any two functional outcome scores assessed by these questionnaires.

In this study, at least three possibilities should be considered for the ineffectiveness of SVF. First, the functional recovery after ACLR is known to be age-dependent. Previous studies have shown that patients aged ≤ 30 years with ACLR have better functional outcomes during the follow-up than patients aged >30 years because of their higher healing potential [38,39]. In previous studies, orthopedic patients who benefited from SVF therapy were either old people with degenerative diseases or had a mean age of over 48 years [20,21,22,23,24,25,26,27,28]. The mean age of the patient cohort in this study was 28 years. Theoretically, SVF should have accelerated healing in younger age groups compared with elderly patients. In fact, in our study, the mean age of the SVF group was 6.3 years older than that of the non-SVF group, and age was significantly associated with repeated-measure functional outcomes, as evidenced by our GEE model analysis. Since the SVF treatment is a self-pay option, the economic factor may contribute to the difference in the mean age of these two study groups. Whether the difference in healing potential because of age difference may offset the possible therapeutical effect in patients with SVF treatment remains unknown. Second, one of the major concerns is the short life span of the biological treatments for augmentation, which limited their efficacy [1]. Thus far, there have been no studies investigating the effective duration of SVF treatment in patients with ACLR. Perhaps, the SVF treatment in our study failed to promote adequate angiogenesis and healing. Third, it is possible that the functional outcome recovery of the ACLR alone (the non-SVF group) was generally excellent partly because of the effect of the endogenous growth factor from bone marrow. This makes it difficult in finding significant differences between the two study groups.

There were some limitations in the current study. First, this is a retrospective study with a small sample size of patients from a single institution. Future prospective investigations with a larger sample size for a longer follow-up duration are warranted. Second, we did not characterize the SVF isolated in this study. However, currently, there is no gold standard protocol for preparing SVF, thereby creating variations in the final product SVF [22]. As such, there are no studies establishing which isolation protocol is better than the other. Third, we did not measure other parameters, such as the radiological outcomes or knee stability, as these were not required to be obtained in our routine follow-up. Future studies should measure both objective outcomes and subjective scores.

## 5. Conclusions

In conclusion, the patients in the SVF group significantly improved all their functional outcomes at 12 months after surgery, but this improvement was not significantly different from those of the patients in the control group. At this point, based on our findings, the use of intraoperative administration of SVF is not justified to improve or accelerate functional recovery after ACLR in young patients. The use of SVF to enhance healing and accelerate recovery after ACLR is still in its infancy. Future research may benefit from standardizing SVF preparation, incorporating multiple doses, and applying this treatment to other age groups.

## Figures and Tables

**Figure 1 jcm-11-06240-f001:**
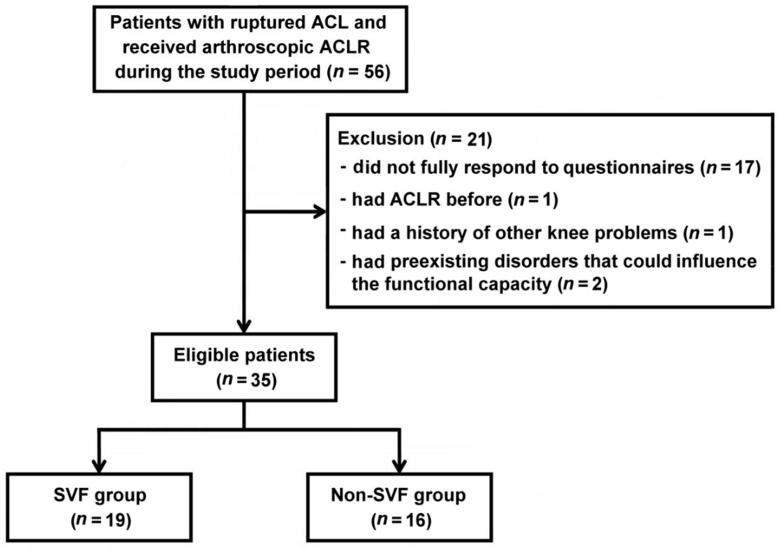
Patient enrollment flowchart: ACL, anterior cruciate ligament; ACLR, anterior cruciate ligament reconstruction; SVF, adipose stromal vascular fraction.

**Figure 2 jcm-11-06240-f002:**
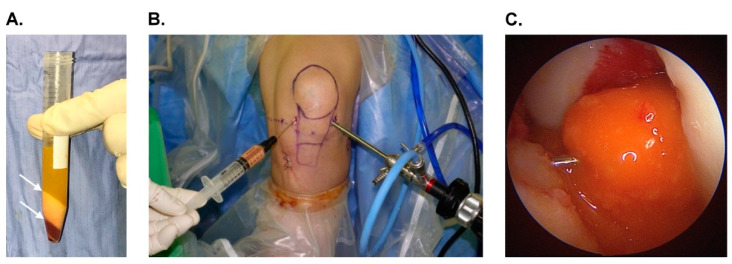
Images showing preparation and administration of adipose stromal vascular fraction (SVF): (**A**) the SVF was the sticky substance under the oil in the tube, which was the yellow layer indicated by two arrows; (**B**) arthroscopic-assisted SVF injection; (**C**) arthroscopic image showing the intrasubstance injection of SVF gel after completing the anterior cruciate ligament reconstruction.

**Figure 3 jcm-11-06240-f003:**
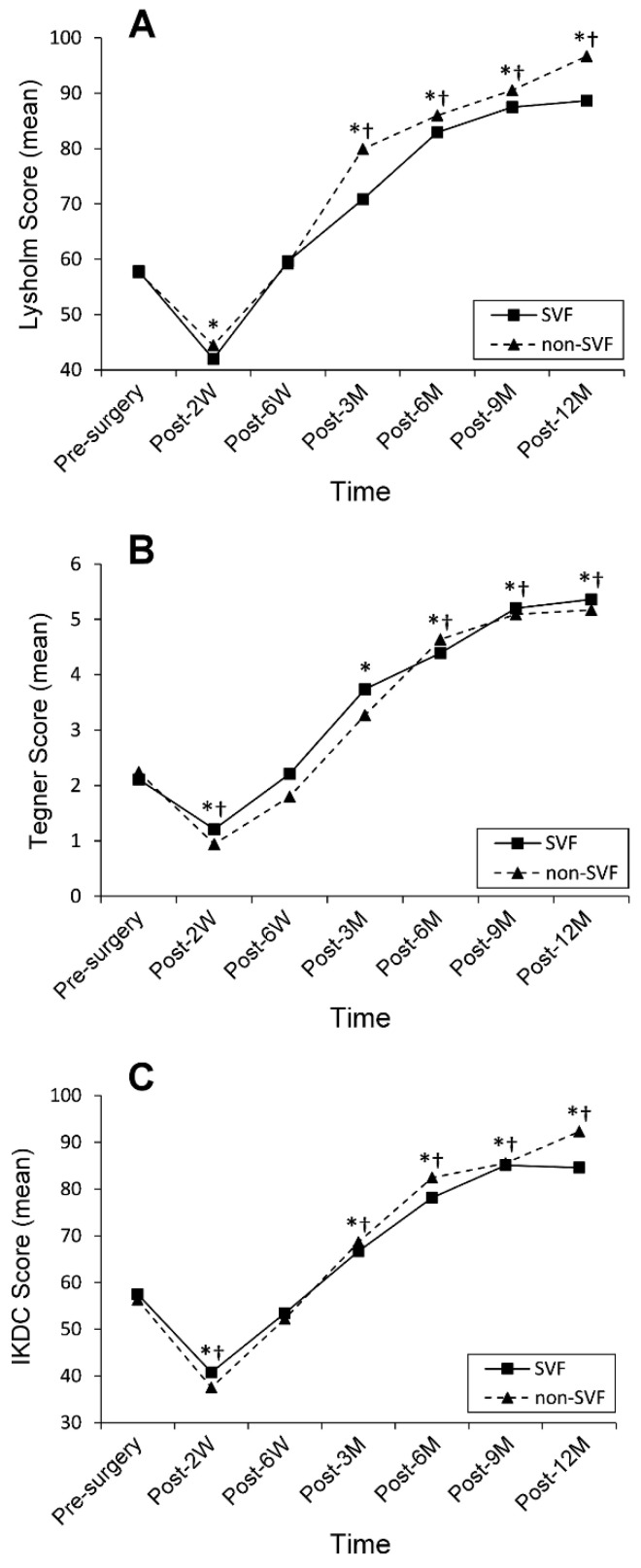
Changes in three types of functional outcome scores in the two study groups during the 12-month follow-up. SVF, adipose stromal vascular fraction; post, post-surgery; W, weeks; M, month; Lysholm, Lysholm knee scoring system; Tegner, Tegner activity scale; IKDC, International Knee Documentation Committee; *, *p*-value < 0.05 compared with pre-surgery scores in the SVF group; †, *p*-value < 0.05 compared with pre-surgery scores in the non-SVF group. Data in the graphs are expressed as absolute mean values.

**Table 1 jcm-11-06240-t001:** Functional outcome scores assessed over time in the two study groups.

	Time	SVF Group (*n* = 19)	Non-SVF group (*n* = 16)	*p*-Value
Lysholm Score	Pre-surgery	57.8 (23.1)	57.6 (28.8)	0.980
	Post-2W	42.0 (19.6)	44.4 (19.4)	0.709
	Post-6W	59.6 (21.6)	59.2 (24.0)	0.956
	Post-3M	70.8 (19.8)	79.9 (18.8)	0.183
	Post-6M	82.9 (14.6)	86.0 (14.2)	0.398^a^
	Post-9M	87.5 (18.6)	90.6 (9.1)	0.838^a^
	Post-12M	88.6 (14.0)	96.7 (5.2)	0.091^a^
Tegner Score	Pre-surgery	2.1 (1.9)	2.4 (1.9)	0.635^a^
	Post-2W	1.2 (0.5)	0.9 (0.4)	0.317^a^
	Post-6W	2.2 (1.0)	1.8 (0.6)	0.228^a^
	Post-3M	3.7 (1.2)	3.3 (1.5)	0.271^a^
	Post-6M	4.4 (1.2)	4.6 (1.4)	0.590
	Post-9M	5.2 (1.5)	5.1 (1.4)	0.850
	Post-12M	5.4 (2.1)	5.2 (1.2)	0.841
IKDC Score	Pre-surgery	57.5 (19.0)	56.3 (22.6)	0.866
	Post-2W	40.8 (10.3)	37.5 (8.6)	0.322
	Post-6W	53.4 (13.0)	52.2 (13.2)	0.791
	Post-3M	66.7 (11.8)	68.7 (14.0)	0.661
	Post-6M	78.1 (12.6)	82.4 (10.4)	0.312
	Post-9M	85.1 (12.9)	85.6 (10.5)	0.919^a^
	Post-12M	84.6 (13.2)	92.3 (9.0)	0.207^a^

SVF, adipose stromal vascular fraction; Lysholm, Lysholm knee scoring system; Tegner, Tegner activity scale; IKDC, International Knee Documentation Committee; post, post-surgery; W, weeks; M, months; ^a^, comparisons were made by Mann–Whitney U test. Other comparisons were made by independent two-sample *t*-test. All data are presented as mean ± SD. A *p*-value < 0.05 was considered statistically significant.

**Table 2 jcm-11-06240-t002:** Correlation matrix of Spearman rank correlation coefficients.

	Lysholm Score	Tegner Score	IKDC Score
Lysholm score	1	-	-
Tegner score	0.736 *	1	-
IKDC score	0.867 *	0.828 *	1

Lysholm, Lysholm knee scoring system; Tegner, Tegner activity scale; IKDC, International Knee Documentation Committee; * a correlation coefficient > 0.7 indicates a strong correlation between two types of scores.

**Table 3 jcm-11-06240-t003:** Multivariate generalized estimating equations analysis of associations between variables and each type of functional outcome scores.

Variables		Functional Outcome Scores					
		Lysholm Score		Tegner Score		IKDC Score	
		β	*p*-Value	β	*p*-Value	β	*p*-Value
SVF							
	non-SVF	Reference		Reference		Reference	
	SVF	2.28	0.553	0.34	0.197	2.12	0.486
Time							
	Pre-surgery	Reference		Reference		Reference	
	Post-12M	**34.29**	**<0.001**	**3.08**	**<0.001**	**30.53**	**<0.001**
Sex							
	Female	Reference		Reference		Reference	
	Male	−0.32	0.937	−0.24	0.393	−1.83	0.566
Age		**−0.84**	**0.001**	**−0.05**	**0.006**	**−0.51**	**0.008**

SVF, adipose stromal vascular fraction; post, post-surgery; M, month; Lysholm, Lysholm knee scoring system; Tegner, Tegner activity scale; IKDC, International Knee Documentation Committee. Multivariate analysis of each variable was adjusted by other confounders. A *p*-value < 0.05 was considered statistically significant, which is indicated by bold font.

## Data Availability

The datasets generated and analyzed during the current study are available from the corresponding author upon reasonable request.

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
