# Peer review of "Intraoperative Administration of Adipose Stromal Vascular Fraction Does Not Improve Functional Outcomes in Young Patients with Anterior Cruciate Ligament Reconstruction"

_jcm, 2022, doi:10.3390/jcm11216240_

Round 1
Reviewer 1 Report
Dear Authors, Thank you for the opportunity to read the content of the manuscript. I find the article interesting. I suggest add the Limitations of the study and comparing the results with the results of other authors in the discussion (2-3 manuscripts). I also believe that the conclusions should refer specifically to the obtained research results. I have no further comments.Author Response
Please see the attachment.

Reviewer 2 Report
Article is written well.
But the methodology section needs more clarification.
Drawing conclusions based only on subjective evaluation forms are prone for bias. Need more objective test results to validate the results and conclusion.
Clarifications are mentioned in the pdf file itself.

Reviewer 3 Report
Without prejudice to the fact that this is a retrospective study, my only concern is with regards to the content of the pre-operative counselling given to these patients. This is important in my opinion since the choice of adipose stromal vascular fraction administration appeared to be an economic one that was thus left to the discretion of the patients. The significant age difference between the study and control groups adds to the curiosity of this finding.
Reviewer 4 Report
Stromal vascular fraction (SVF) - no clinical studies investigating the benefits of augmentation of the anterior cruciate ligament reconstruction (ACLR) with SVF. The retrospective study in assessing the effects of intraoperative SVF administration on the functional outcomes in young patients with ACLR show that intraoperative administration of SVF does not improve or accelerate the functional recovery after ACLR in young patients.
